# Characterization of patients requiring inpatient hospital ethics consults- A single center study

**Aditya Mahadevan**[1]☉, **Armon Azizi**[1]☉, **Cyrus Dastur**[2], **Sara Stern-Nezer**[2], **Jeffry Nahmias**[3], **Farshid Dayyani**[4]*

**1** University of California Irvine School of Medicine, Irvine, Irvine, CA, United States of America, **2** Department of Neurology, Division of Neurocritical Care, University of California, Irvine, Irvine, CA, United States of America, **3** Department of Surgery, Division of Trauma, Burns, Critical Care & Acute Care Surgery, University of California, Irvine, Irvine, CA, United States of America, **4** Department of Medicine, Division of Hematology and Oncology, University of California, Irvine, Irvine, CA, United States of America

☉ These authors contributed equally to this work.

* fdayyani@hs.uci.edu

**Data Availability Statement:** All relevant data are within the paper and its Supporting Information files.

## Abstract

### Introduction

Ethics consultations are often needed at difficult junctures of medical care. However, data on the nature of how patient characteristics, including race/ethnicity, language, and diagnosis, affect ethics consult outcomes are lacking.

### Methods

We performed a retrospective cohort study of all patients who were seen by the Ethics Consult Service between 2017 and 2021 at a large tertiary academic center with the aim of determining whether patient demographic and clinical factors were associated with the timing of ethics consult requests and recommendations of the ethics team.

### Results

We found that patients admitted for COVID-19 had significantly longer median times to consult from admission compared with other primary diagnoses (19 vs 8 days respectively, p = 0.015). Spanish-speaking patients had longer median times to consult from admission compared to English speaking patients (20 vs 7 days respectively, p = 0.008), indicating that language barriers may play a role in the timing of ethics consultation.

### Conclusions

This study demonstrates the need to consider clinical and demographic features when planning and prioritizing ethics consultations at large institutions to enhance consult efficiency, resource utilization, and patient experience and autonomy.

**Funding:** The author(s) received no specific funding for this work.

**Competing interests:** The authors have declared that no competing interests exist.

## Introduction

Ethics consults serve to support informed decision-making between patients, their families, and the healthcare team. Ethics consultations are requested for several reasons, including conflict resolution, surrogate decision maker determination, goals of care (including end-of-life care), and patients' emotional triggers [1–4]. Less commonly, religious and cultural issues may prompt ethics consultations [1]. Whether the frequency of consultations and ethics team recommendations are associated with patient demographics and characteristics, like admission diagnosis, language, and religion is unknown.

Patient characteristics, notably race and gender, have been shown to be correlated with the duration of ethics consultations and time from admission to consult request, with women receiving ethics consults earlier in hospitalization, and African-Americans receiving ethics consultations later than Caucasian patients [5]. Although the association between ethics consultation and certain patient characteristics has been studied, the role of language and primary diagnosis in ethics consultation outcomes has still not been explored in depth despite these characteristics affecting outcomes such as clinical course and transition to end of life care [6, 7]. Prior works have summarized ethics consult data and outcomes and explored the relationship between patient characteristics and family satisfaction with ethics consultation [5–8]. However, to the best of our knowledge, the association between patient language spoken, and ethics consult timing center-wide as well as between different diagnoses has not been evaluated. Therefore, this study aimed to describe the associations between clinical and demographic patient characteristics such as language and primary diagnosis and the outcomes of ethics consultations, namely the timing of the ethics consultation from admission and the set of recommendations made by the ethics team.

Given that over 25 million Americans are limited English proficient, it follows that a high proportion of admitted patients are non-English speaking, highlighting the importance of investigating the role of language in ethics consultation [9]. As limited English proficiency may result in poor patient comprehension, we hypothesized that non-English language speakers would experience increased times from admission to ethics consultation due to challenges stemming from patient-provider communication and health literacy [10, 11]. Furthermore, given the role of diagnosis on illness acuity and hospital course, we hypothesized that patients diagnosed with COVID-19 or cancer would have longer time to ethics consultation, compared to other diagnoses, given that consultations for these patients would be centered around topics related to end of life care, resulting in the initiation of consultation at later times.

## Materials and methods

A retrospective chart review was performed using data from patients treated at a single, large tertiary care academic center (University of California Irvine Medical Center) from 2017 to 2021 who received an ethics consult. The study was exploratory, and patients were not registered prior to the analysis. This study was designated as IRB exempt by the University of California Irvine IRB and thus a waiver of consent granted.

### Cohort identification

All inpatient ethics consultations performed by the University of California, Irvine Health Ethics Consultation Service during the study years were included. Ethics consultations at UC Irvine were initiated by the treatment team and cases were reviewed by the ethics consult team, comprised of multiple physicians including a Medical Director with additional ethics training and expertise. In addition, the teams included members from nursing, legal/risk management and spiritual services. Not all ethics consults necessitated an in-person disciplinary

meeting with the patient or family. In most cases, a meeting was held between the primary care team and ethics team to discuss recommendations within 24 hours of consultation and the ethics team met with the patient on an as-needed basis. Consultations were performed as needed, meaning that urgent consultations may have been performed at night or on weekends.

## Chart review

Demographic and clinical data were extracted by chart review. The primary outcome was the time from admission to ethics consultation. The secondary outcome was the set of recommendations made by the ethics consult team, which included transition to comfort care (i.e. removal from life support including mechanical ventilation), no escalation of care, option to transfer of care to an outside facility either directly or via a transfer letter, and designating someone as do-not-resuscitate (DNR), among other recommendations. The recorded data included age, sex, principal medical diagnosis necessitating the consult (as documented in the consult note), time from admission to first ethics consultation, and number of ethics consultation encounters during admission. Demographics were defined based on self-reported patient information that was obtained during admission or prior outpatient visits at the institution. Primary diagnosis was defined as the primary reason for admission. These were obtained through chart review as ICD codes vary between provider and were inconsistent between patient groups. In addition, information regarding the patient included the primary language spoken, religion, decision making capacity (as determined by the ethics consult team), location of the patient within the hospital at the time of consult (Floor, intensive care unit, and inpatient psychiatry unit), and whether the patient died during admission. Language fluency was determined by the patient's reported preferred language.

## Statistical methods

Patients were stratified by language spoken, principal diagnosis, decision-making capacity, hospital location at the time of ethics consult, and mortality. A cox proportional hazards model was used to compare times to consult between English speakers and either Spanish or Vietnamese speakers. Features to be included in the multivariate model were selected based on univariate statistical testing differences as determined by standardized mean difference (SMD) and significant p-value (**S1 Table**). A SMD cutoff of greater than 0.8 and adjusted p-value of $< 0.05$ was utilized to deem variables as significantly different between demographic groups. Two-sided student t-tests were used to compare time to consult between different diagnosis groups. Chi-squared tests were used to compare ethics team recommendations between patient groups stratified by either primary diagnosis, admission location, or mortality during admission. The significance threshold was set at 0.05 for all analyses. All statistical analyses were performed in R version 4.0.5 using the *survminer*, *survival*, and *stats* packages [12–14].

## Results

A total of 236 patients were identified and included in the study. The cohort was 33.9% female sex and had a median age of 63 (range 0.1–102 years). In terms of primary language spoken, 70.3% were English-speaking, 12.7% Spanish-speaking, and 10.2% Vietnamese-speaking. In terms of religious affiliation, the large self-identified group identified as Christian (44%) followed by Buddhist (3.4%) and Muslim (1.4%), however, the religion of 45.8% of patients was unknown (**Table 1**).

The most common major diagnoses upon admission among the cohort were cancer (18%), infection (15%), and neurologic diagnoses (12%) (**Table 1**). In total, 84% of patients lacked capacity due to intubation, unconsciousness, psychiatric illness, or other reasons, which was often the primary reason for ethics consultation. The location of admission across the cohort

**Table 1. Overall demographics and characteristics of the study population.**

|  | N | % |
|---|---|---|
| **Median Age (range)** | 63 years (0.1–102) | |
| **Sex** | | |
| Female | 80 | 33.9 |
| Male | 156 | 66.1 |
| **Language** | | |
| English | 166 | 70.3 |
| Spanish | 30 | 12.7 |
| Vietnamese | 24 | 10.2 |
| Korean | 4 | 1.7 |
| Mandarin | 1 | 0.4 |
| Other | 12 | 5.1 |
| **Religion** | | |
| Christian | 104 | 44.1 |
| Buddhist | 8 | 3.4 |
| Muslim | 4 | 1.7 |
| Jewish | 1 | 0.4 |
| Other | 11 | 4.7 |
| Unknown | 108 | 45.8 |
| **Primary Admission Diagnosis Classifications** | | |
| Cancer | 45 | 19.1 |
| Covid19 | 18 | 7.6 |
| Other Infection | 36 | 15.3 |
| Drug Abuse | 5 | 2.1 |
| Psychiatric | 40 | 16.9 |
| OB/GYN | 10 | 4.2 |
| Neuro | 39 | 16.5 |
| Cardiac | 22 | 9.3 |
| Trauma | 19 | 8.1 |
| Other | 146 | 61.9 |
| **Capacity** | | |
| Yes | 28 | 11.8 |
| No | 199 | 84.3 |
| **Hospital Location** | | |
| ICU | 74 | 31.4 |
| Floor | 73 | 30.9 |
| Neurology | 23 | 9.7 |
| Psychiatry | 10 | 4.2 |
| Other | 42 | 17.8 |

Abbreviations: OB/GYN—Obstetrics and Gynecology; ICU–Intensive Care Unit

varied with most patients being admitted to the Intensive Care Unit (ICU) or floor. The reasons for ethics consultation varied widely across all patients. Some of the reasons for ethics consultation in our cohort included assisting with family discussions about patient care, determining patient's surrogate decision maker, determining a patient's candidacy for cesarian section, and assessing the benefit and appropriateness of a tracheostomy, among others.

Patients had a median of one ethics consult during their admission (median: 1, range: 1–6) and 44.9% of patients died during index hospitalization. Regarding ethics consult

**Table 2. Mortality and ethics recommendations.** Ethics recommendation percentages indicate the percentage of patients who received specific recommendations and are not exclusive (each consultation had one or more associated recommendations, therefore, percentages may sum to greater than 100%).

|  | N | % |
|---|---|---|
| **Median Time To Consult (range)** | 8.5 days (0–83) | |
| **Death during admission** | | |
| Yes | 106 | 44.9 |
| No | 116 | 49.2 |
| Unknown | 10 | 4.2 |
| **Ethics Recommendation** | | |
| Pursue Comfort Care | 72 | 30.5 |
| No escalation of care | 61 | 25.8 |
| Transfer/Letter | 27 | 11.4 |
| Make DNR | 76 | 32.2 |
| Other | 134 | 56.8 |

Abbreviations: DNR–Do Not Resuscitate

Transfer/Letter: recommendation to transfer medical care to another institution or a transfer letter requesting further care at an alternative healthcare location

recommendations, 32.2% of consults recommended it was ethically appropriate to transition the patient to DNR designation, 30.5% recommended transition to comfort care, and 25.8% recommended no escalation of care (**Table 2**).

## Ethics consultation patterns based on patients' primary language spoken and diagnosis

English speaking patients received ethics consultations a median of seven days after admission, in contrast to Spanish-speaking and Vietnamese-speaking patients, who received consultations a median of 20 and 12 days into hospital admission respectively, (**Fig 1**). Spanish-speaking patients received consults significantly later in their admission than English-speaking patients (p = 0.008), when controlled for age (**Fig 1**). There was no significant difference in time to consult between English-speaking patients and Vietnamese-speaking patients (p = 0.315). There were no significant differences in clinical or demographic characteristics between English speaking and Spanish-speaking patients (**S1 Table**). Vietnamese speaking patients were significantly older and had a different distribution of religious affiliation compared to English speaking patients (**S1 Table**).

## Ethics consultation patterns in patients with COVID-19

COVID-19 had the longest median time from admission to consult at 17.5 days (range 4–53), followed by Obstetrics & Gynecology (OBGYN) and cancer related cases at 15.5 (range 0–35) and 12.5 (range 0–53) days respectively. Cardiac, trauma, and neurologic cases had the shortest median time to consult at 5.5 (range 0–49), 7 (range 1–37), and 7 (range 1–48) days respectively (**Fig 2**). When compared to all other patients, COVID-19 patients had a significantly longer time to consult in comparison to other diagnoses (p = 0.015).

## Ethics consultation patterns based on patient's primary diagnosis

When evaluating the association between primary diagnosis and number of ethics consultations received, there were no significant differences observed in number of ethics consults

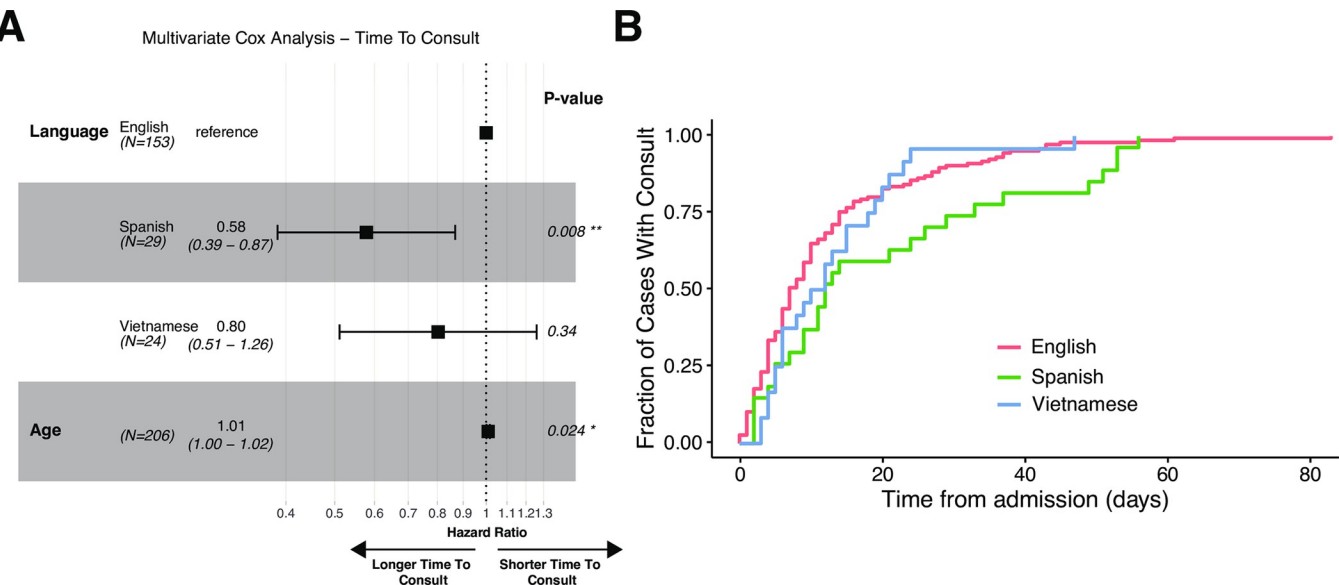

**Fig 1. Multivariate cox analysis of differences in the time to consult from admission between different language groups.** A) Forrest plot depicting the effect size of language on time from admission to consult (English vs Spanish and English vs Vietnamese) when controlled for Age. B) Kaplan Meier curve showing the accumulation of ethics consults over time in different language groups.

between diagnosis groups. There was however, a low amount of variability in this data since almost all patients had only one ethics consult during admission (**S1 Fig**).

### Differences in ethics team recommendations between diagnosis, admission location, and mortality groups

When evaluating the association between primary admission diagnosis and ethics consult recommendations, COVID-19 had numerically the highest proportion of recommendations to designate patients as DNR, followed by cardiovascular diagnoses, while OBGYN diagnoses had the lowest rates. Trauma had the highest proportion of recommendations to pursue comfort care, followed by cancer and COVID-19, while OBGYN had the lowest rates, however, none of these differences were statistically significant (**Fig 3A**, **S2 Table**).

Compared to patients with capacity, patients lacking capacity did not have significant differences in their recommendations to receive the DNR designation, to not escalate care, to transfer care to another facility, or pursue comfort care (p = 0.13) (**Fig 3B**). There were, however, differences in ethics recommendation outcomes between different admission locations within the hospital. Specifically, patients in the ICU had higher rates of recommendations to pursue comfort care and transfer to hospice compared with patients admitted to locations elsewhere in the hospital (p<0.01) (**Fig 3C**). Additionally, there was a significant association between ethics consult recommendations and patient mortality during admission. Patients who died during admission following ethics consultation had higher rates of recommendations to not escalate care, make DNR, pursue comfort care, and transfer to hospice relative to patients who ultimately survived (p<0.0001, **Fig 3D**).

### Discussion

To our knowledge, this is the first study to evaluate the association between patient language spoken, primary clinical diagnosis, and inpatient ethics consultation outcomes over a 5-year

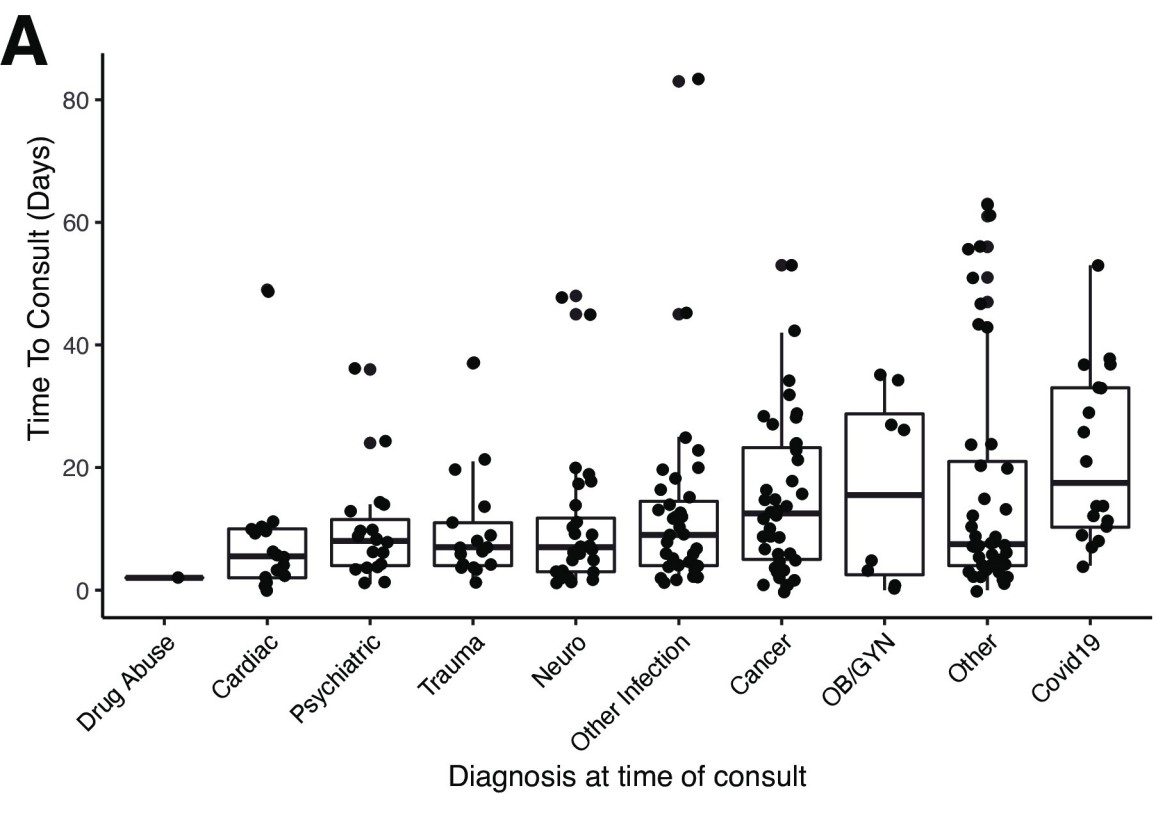

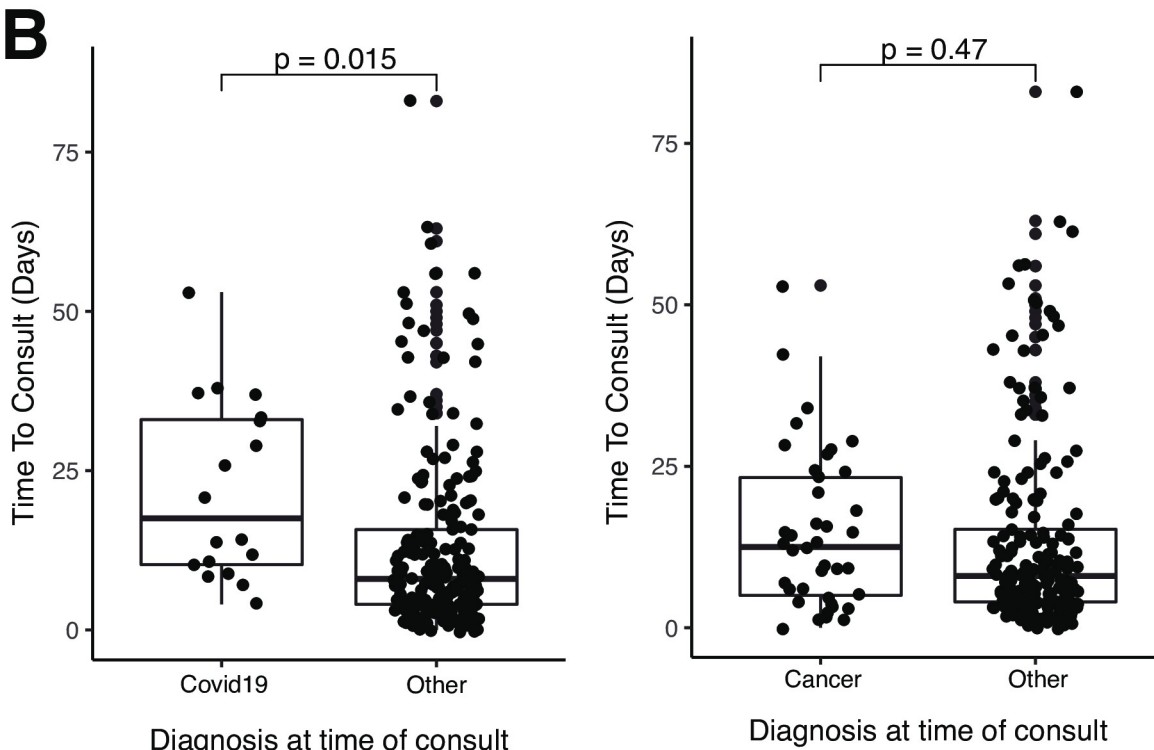

**Fig 2. Time from admission to consult in different diagnosis groups.** A) Time to consult across patients with different admission diagnoses. B) Time to consult between patients with a diagnosis of covid and all other patients (Student t-test, p<0.05). C) Time to consult between patients with a diagnosis of cancer and all other patients (Student t-test).

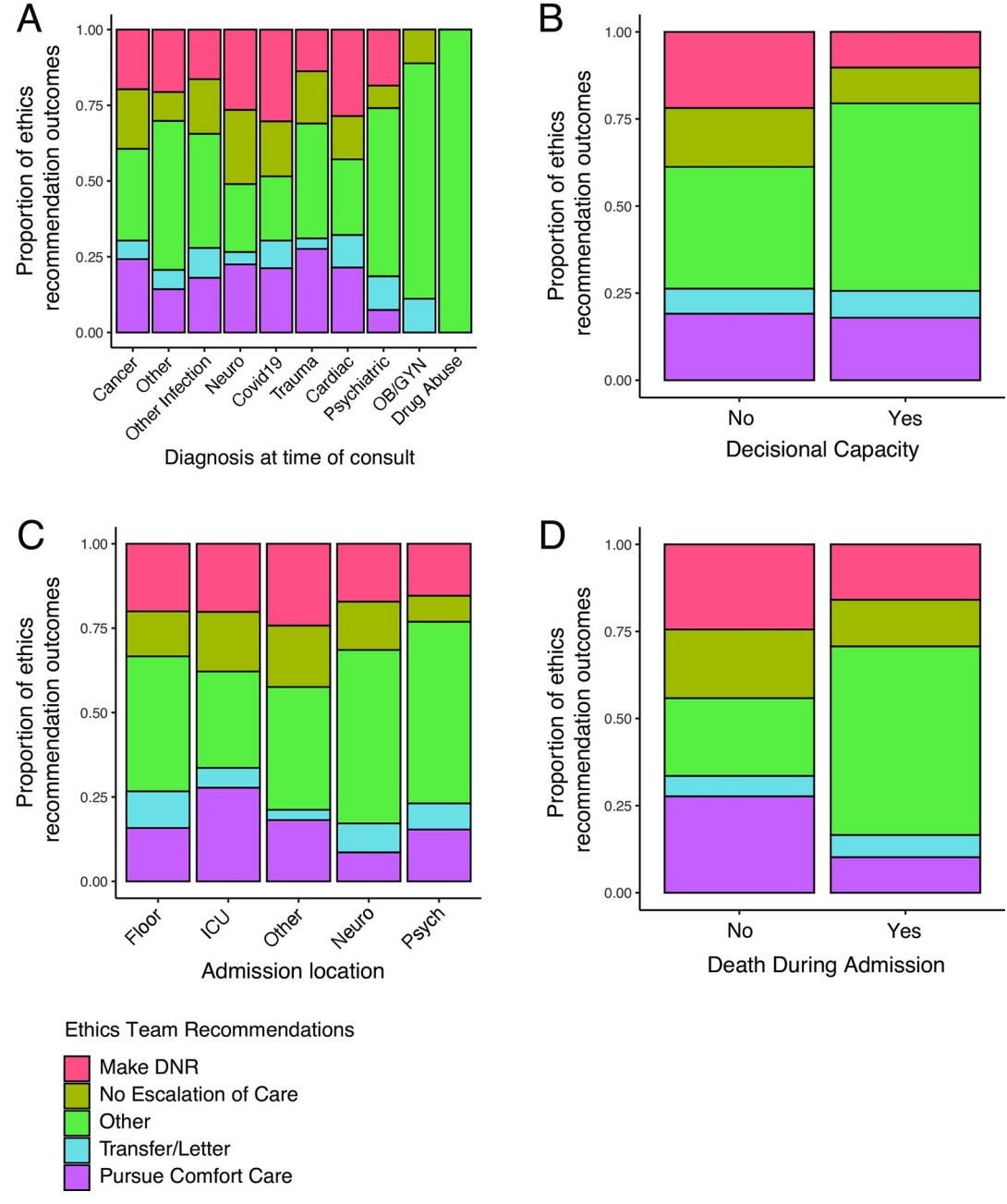

**Fig 3. Differences in ethics team recommendations between diagnosis, admission location, and mortality groups.** A) Ethics recommendation distribution based on admission diagnosis. B) Ethics recommendations based on patient capacity. C) Ethics recommendations based on admission location within the medical center. D) Ethics recommendation association with mortality during admission. Note that ethics recommendations are not exclusive, and each consultation had one or more recommendations.

period. We found that Spanish-speaking patients and patients with COVID-19 had ethics consults significantly later in hospital course, and that decision-making capacity had no significant association with committee recommendations. Better understanding the contributions of patient language spoken and primary diagnosis to ethics consultation can impact how and when ethics consultations are requested.

Our cohort was primarily English-speaking, male, and Christian. However, a significant proportion of our patients spoke other languages, or practiced other religions. Per published population data, 45% of Orange County, California residents (the primary catchment area for UC Irvine Medical Center) speak a language other than English at home [15]. However, this data does not differentiate between speaking English at home and English proficiency (i.e. not requiring an English interpreter), which may explain this discrepancy. As such, our cohort was representative of the patient population of Orange County.

In our 5-year cohort, the most common admission diagnoses in patients receiving ethics consultations were cancer, psychiatric complaints, and neurologic complaints–consistent with prior published data [16]. However, COVID-19 accounted for 18.2% of our consults from 2020–2021, which made it the most common diagnosis eliciting an ethics consult during those years. This was substantially lower than the 44.9% rate found during a study conducted at the onset of the COVID-19 pandemic (March–June 2020) [17].

When controlling for age, Spanish-speaking patients had ethics consultations at a significantly later date in their hospital stay when compared to English-speaking patients. Prior literature has revealed substantial differences in time to ethics consultation based on race and gender, but not primary language [5]. Our findings were unique to Spanish-speaking patients, as Vietnamese-speaking patients did not have significant discrepancies in time to ethics consultation. Varied perception of disease states, which differ by culture/across ethnolinguistic groups, may contribute to these discrepancies in time to consultation [18]. Taken together, our findings suggest that ethnolinguistic barriers may contribute to delays in ethics consultation. This is of particular concern given that delays in obtaining ethics consultation may harm patient autonomy and place unnecessary burden on the healthcare system [19, 20]. While the role that patient characteristics (e.g., race, gender) play in ethics consultations has been explored previously, additional research is needed to better understand the impact of spoken language and cultural context on the timing of ethics consultations [5].

Cancer and COVID-19 had significantly longer times to consult than other diagnoses, consistent with the hypothesis that consults for these diagnoses were centered around end-of-life care due to the often terminal nature of cancer and the drawn-out hospital courses of patients with COVID-19. This was consistent with our data demonstrating that ethics consults for patients with COVID-19 diagnoses had among the highest rates of recommendations to make DNR or transition to comfort care, consistent with a prior study highlighting ethics consults related to end-of-life care in patients with COVID-19 [17]. However, more data is needed to draw significant conclusions on causal relationships between principal diagnosis and time to ethics consultation.

In our cohort, the recommendations given by the ethics committee varied between different diagnostic groups. There were higher rates of recommendations to make the patient DNR, not escalate care, or withdraw care in patients with COVID, Cardiac, Neurological, and Cancer diagnoses relative to other groups. Our data showed that certain diagnoses were associated with more "drastic" recommendations such as terminating care or withdrawing care. Our data revealed that patients who received recommendations of DNR, no escalation of care, or to pursue comfort care had significantly higher mortality rates during their admission. Certain diagnostic groups, such as OBGYN and psychiatric, were more often associated with the "Other" category of recommendations, which included conflict resolution, planning with patient families, continued evaluation of patient capacity, and identifying appropriate surrogate decision makers. Further studies are needed to better understand the implications of ethics consult recommendations in the context of different disease pathologies and how these recommendations are related to overall patient outcomes.

Better understanding the contributions of patient characteristics, like primary language spoken, and clinical factors, like primary diagnosis, to ethics consultation can impact how and

when ethics consultations are requested. By identifying differences between ethics consultations in different disease contexts, we can better understand which cases are more likely to benefit from ethical consultation and are more likely to have medical decisions changed following an ethics consultation. Given that ethics consultations are often requested directly by treatment teams, incorporating these insights could lead to changes in ethics consult resource dedication and patient prioritization in large institution hospital settings. In turn, this could improve the efficiency with which certain patients can be seen by ethics teams and the effectiveness of ethics recommendations and outcomes. Prior literature suggests that usage of ethics consultations is associated with lower resource expenditure compared to those that did not receive them; improving efficiency and timing of ethics consultation would likely compound this benefit [21].

This study had a number of limitations. As a retrospective chart review, we are unable to establish causality/directionality in our associations. Furthermore, regarding our language/ religion analyses, language and religion are not always kept up to date, or accurately recorded in the chart. As such, we were unable to verify or quantify the usage of translators as these were not consistently documented in the chart. Additionally, in patients without capacity, we were unable to verify whether the language the patient spoke was consistent with that of their surrogate decision-maker. While we associated our ethics consultations with a single (primary) diagnosis, a single diagnosis cannot capture the complexity of a patient's medical course, or more nuanced recommendations, like searching for a surrogate decision-maker, or simply engaging in a goals of care discussion with the family. Furthermore, certain categories of diagnoses are inherently more common in certain demographics, for example, obstetric diagnoses in women of childbearing age which may lead to biases in the utilization of ethics consults. Additionally, clinical and demographic factors that were not evaluated in our chart review may function as confounders or mediators in the relationships that were observed in our data and should be taken into consideration in the interpretation of our results. Nevertheless, by performing a detailed chart review and using multivariate analyses, we aimed to minimize the impact of these limitations on the interpretation of our results.

## Conclusions

Our study is the first to comprehensively document ethics consults over a 5-year period, including time to consult, recommendations, outcomes, and corresponding patient characteristics in a large and diverse cohort of patients. Of note, we found that patients admitted for COVID-19 had significantly longer median times to consult from admission and Spanish-speaking patients had longer median times to consult from admission compared to English speaking patients. These data and associated analyses allow for a better understanding of the resource management and timeliness required to provide ethics consultations to patients of diverse backgrounds, beliefs, and clinical circumstances. While taking a case-by case approach to medical ethics consultation ensures that patient autonomy and wishes are preserved, the contribution of patient characteristics, diagnosis, and clinical acuity to ethics consultation timing and recommendations cannot be ignored. Overall, this study demonstrates the need to further explore the role of clinical and demographic features in planning and prioritizing clinical ethics consultations in inpatient settings in order to ultimately improve consult efficiency and enhance patient experience and autonomy.

## Supporting information

**S1 Checklist.**
(DOCX)

**S1 Data.**
(XLSX)

**S1 Fig. Number of ethics meetings for patients in each admission diagnosis group.**
(DOCX)

**S1 Table. Demographic and clinical data in different language groups.** P values indicate significant differences in demographic data between groups.
(DOCX)

**S2 Table. Percentage of ethics recommendation outcomes in each diagnostic group.** Sum of percentages may be greater than 100% for each diagnosis since patients receive multiple recommendations.
(DOCX)

## Acknowledgments

We would like to thank Sheila Anaya for her assistance in identifying the consults that were performed at UCI during the study period.

## Author Contributions

**Conceptualization:** Aditya Mahadevan, Armon Azizi, Farshid Dayyani.

**Data curation:** Aditya Mahadevan, Armon Azizi, Cyrus Dastur, Sara Stern-Nezer, Jeffry Nahmias, Farshid Dayyani.

**Formal analysis:** Armon Azizi.

**Investigation:** Aditya Mahadevan, Armon Azizi, Farshid Dayyani.

**Methodology:** Aditya Mahadevan, Armon Azizi.

**Project administration:** Cyrus Dastur, Farshid Dayyani.

**Software:** Armon Azizi.

**Supervision:** Farshid Dayyani.

**Validation:** Aditya Mahadevan, Armon Azizi.

**Visualization:** Armon Azizi.

**Writing – original draft:** Aditya Mahadevan, Armon Azizi, Cyrus Dastur, Sara Stern-Nezer, Jeffry Nahmias, Farshid Dayyani.

**Writing – review & editing:** Aditya Mahadevan, Armon Azizi, Cyrus Dastur, Sara Stern-Nezer, Jeffry Nahmias, Farshid Dayyani.

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
