## [Decision Letter · Decision Letter 0]

26 Oct 2023

PONE-D-23-13646Characterization of Patients Requiring Inpatient Hospital Ethics Consults- A Single Center Study

PLOS ONE

Dear Dr. Dayyani,

Thank you for submitting your manuscript to PLOS ONE. After careful consideration, we feel that it has merit but does not fully meet PLOS ONE’s publication criteria as it currently stands. Therefore, we invite you to submit a revised version of the manuscript that addresses the points raised during the review process.

1.Kindly specify what your study's aims and objectives are? Also if there is any hypotheses which were to be tested by this study, kindly clarify these in your Introduction. For example "you identified multiple knowledge gaps (1. How frequency of ethics consultations and ethics team recommendations are associated with patient demographics and 2. How role of language and primary diagnosis are associated with patient outcomes). I would recommend making the knowledge gap concise and clear to identify the reason for your study."

2. Further, ' this study aimed to describe the associations between clinical and demographic patient characteristics such as language and primary diagnosis and the outcomes and recommendations of ethics consultations. I would recommend describing what you mean by outcomes. Are these patient outcomes like transition to end of life care?'

3.  Kindly explain why you "conducted Cox regression hazard model using the age as covariate? Could and should other factors be included? If the authors want to justify the use of age as the sole covariate, they should explain the reason why."

4. Kindly elaborate and provide more information about how the ethics consultations are conducted at the institution where this study was conducted. Does it differ in any way from the practice at other institutions as reported in the literature?.

5. Kindly clarify your study outcomes, e.g. time to, or frequency of ethics consults, or impact of language barriers, etc...

6. How were 'patient demographics defined. Were patients asked to self-identify race, sex, religion etc.'

7. Also, kindly specify how you determined primary diagnosis? Was it by ICD code for hospital admission? If so please include this in the methods section.

8.  Also kindly elaborate on the language skills and use of language aids, such as smart translators, etc.

9. Kindly streamline your discussion section to focus more on fewer relevant topics consistent with findings from your own study's objectives.

10. Kindly include data on reasons for ethics consults in your results section. 

11. Kindly explain the meaning of "transfer/letter" in Table 2.

12. In Figure 1A and Figure 2 kindly add an x-axis, and label P=values for better clarity for readers.

13.  Kindly summarize the most important findings from your study in the conclusions and include these in the abstract section.

14. Kindly address all other comments and observations as raised by the peer-reviewers.

15. Please correct all typographical and grammatical errors in your manuscript before resubmission.

We look forward to receiving your revised manuscript.

Kind regards,

Sylvester Chidi Chima, M.D., L.L.M

Academic Editor

PLOS ONE

2. PLOS requires an ORCID iD for the corresponding author in Editorial Manager on papers submitted after December 6th, 2016. Please ensure that you have an ORCID iD and that it is validated in Editorial Manager. To do this, go to ‘Update my Information’ (in the upper left-hand corner of the main menu), and click on the Fetch/Validate link next to the ORCID field. This will take you to the ORCID site and allow you to create a new iD or authenticate a pre-existing iD in Editorial Manager. Please see the following video for instructions on linking an ORCID iD to your Editorial Manager account: https://www.youtube.com/watch?v=_xcclfuvtxQ.

**Comments to the Author**

1. Is the manuscript technically sound, and do the data support the conclusions?

Reviewer #1: No

Reviewer #2: Partly

2. Has the statistical analysis been performed appropriately and rigorously? 

Reviewer #1: No

Reviewer #2: Yes

3. Have the authors made all data underlying the findings in their manuscript fully available?

Reviewer #1: Yes

Reviewer #2: No

4. Is the manuscript presented in an intelligible fashion and written in standard English?

Reviewer #1: Yes

Reviewer #2: Yes

5. Review Comments to the Author

Reviewer #1: A very interesting topic on the role of the ethics committee in the era of COVID-19. I particularly found the result interesting, which showed the more time spent in getting the ethics committee consultation among Spanish speaking patients, not Vietnamese speaking. However, the study design and the methods needed improvement, which made this study difficult to accept to the journal.

Most importantly, I am not certain about the main research question. If it was about the language, they should have focused on it and discuss the issue suggested below. If it was about COVID-19, they should have done it likewise. The study was so vague in designing, and I found the same in the discussion part, which became irrelevant to their findings frequently.

Major comments

I am not sure why the authors conducted Cox regression hazard model using the age as covariate? Could and should other factors be included? If the authors want to justify the use of age as the sole covariate, they should explain the reason why.

The language skills or the level of language aids, such as smart translators, should be clearly documented. Were Spanish speaking patients able to discuss with the team including Spanish speakers? How often human/AI translators were available and used?

I am interested in the decision making process on the patients who lost the capacity. Was the decision made based on the patients' advance care directives, or the will of the family? If family members were involved, what was their primary language to speak?

From clinical point of view, I am not surprised to see that it took long time to have ethical consultation for COVID-19 patients, who typically have severe respiratory failure, yet the rest of organs are relatively spared, hence it takes for a long time to judge that the patients had doomed prognosis (if they had).

Minor comments

Tense of the sentences should be edited. For example I noticed that something happened in the past was described in the present tense.

Reviewer #2: This study by Mahadevan et. al. quantitatively assessed ethics consults at one academic medical center over a 5 year period to assess the association of patient demographic and clinical factors with ethics consults, specifically the timing of the consults and recommendations by the ethics team. They found differences in timing of consults, with patients with COVID-19 and Spanish speaking having longer time from admission to consult.

I think the data in this study has promise, but there are limitations in how this study is described. After reading the article multiple times, I am still unclear as to what the primary and secondary aims are and the specific limitation in prior knowledge this this study is filling. The lack of inclusion of the reason for ethics consult is a major limitation of this paper, as this reason would also be associated with patient outcomes. The discussion includes multiple topics that are briefly discussions and would benefit from focusing on a few topics with more detail.

Introduction:

I had a hard time understanding exactly what you aim to assess in the manuscript as you identified multiple knowledge gaps (1. How frequency of ethics consultations and ethics team recommendations are associated with patient demographics and 2. How role of language and primary diagnosis are associated with patient outcomes). I would recommend making the knowledge gap concise in clear to identify the reason for your study.

In your aim statement, Therefore, this study aimed to describe the associations between clinical and demographic patient characteristics such as language and primary diagnosis and the outcomes and recommendations of ethics consultations would recommend describing what you mean by outcomes. Are these patient outcomes like transition to end of life care?

In the last sentence of the introduction, The high proportion of non-English speaking patients across healthcare systems nationally highlights the importance of investigating the role of patient characteristics that may drive disparities in clinical care. Why would you expect non-English speaking patients to have increased time to ethics consult? I would recommend quantifying here how many patients across the US have a high proportion of non-English speakers and specifically identify why you believe this group would experience disparities in time to ethics consult.

For your hypothesis, given the role of diagnosis on illness acuity and hospital course, we hypothesized that there would be differences in ethics consult timing and recommendations by patient diagnosis. What do you think this difference will be? Will more acute illness be associated with sooner consults or vice versa? Recommend describing your thought process here rather than just say differences exist.

Methods:

Under Cohort Identification, I would recommend adding more description about how the ethics consult occurs. Do all the ethics team members meet with the patient within 24 hours of a consult? Do all consults require a meeting in person with the patient? People reading this article from other centers might be surprised that this is what ethics teams do at your center because at other centers ethics teams do not see every patient. So it would be great to have more detail here so readers can identify how your ethics team structure may be similar or different than their structure.

Under Chart Review:

You state your primary outcome was time from admission to ethics consultation. However you identified in your introduction that demographic information about ethics consults already exist with women receiving earlier consults during the hospitalization. I think you need to make it very clear in this manuscript how your study is different than prior studies. Is it because you are only looking at the association of language and patient diagnosis with your primary outcome of time from admission to ethics consultation?

If you plan for one of your outcomes to focus on assessing frequency of ethics consults, I would specifically identify this. You mention you collected this data in the methods and in the introduction identify this frequency of consults as a knowledge gap, but then in your aims you do not specifically describe frequency of consults as outcome you are looking for.

I would recommend adding how patient demographics were defined. Were patients asked to self-identify race, sex, religion etc. asked on hospital admission? Did admission staff make an opinion about someone’s race and add that to their medical record?

How did you determine primary diagnosis? Was it by ICD code for hospital admission? Recommend including that in methods.

Results:

Do you have data on the reason for ethics consult? You describe the recommendation for these consults, but if the consult was only for help with determining a surrogate, then a change in patient outcome may not occur. It would be great to see the reason for consult and consider breaking down patient outcomes by the reason for consult.

In Table 2, what does Transfer/Letter mean? Recommend adding more detail here. Also for Table 2, can patients be counted multiple times under the Ethics Recommendation category? I would recommend making that clear here as the percentages do not add to 100.

For Figure 1 A, recommend adding to x axis what the numbers represent. Recommend adding to values in right column whether they are p-values, and what asterisks represent.

For Figure 2, is the 0.015 and 0.47 at the top p values? If so, recommend including that in Figure legend. You write in the legend about t-tests, but it is unclear whether they correlate with those two values listed on the table.

Discussion:

I think it would be helpful if you defined exactly what you think the most important findings are from your research at the beginning of the discussion, and then clearly describe how your findings add to what is already known in the literature. In your first sentence, assessing the association between patient characteristics and inpatient ethics consults has been done before. It is important that you clearly state how your study is different (that you looked specifically at language and patient diagnosis). If number of ethics consults is an important part of your results that you are going to include in your discussion, it would be helpful to state that data in your summary of the discussion in your first paragraph of the discussion.

There are a lot of topics being presented in the discussion, without much detail for individual topics. I would recommend picking a few findings (2-3) of interest and then going into more detail about what these findings mean. For example, if you want to discuss Spanish speaking patients getting later consults than other patients, you can describe more fully why that difference exists and what the importance of that difference is than what is currently included in that section. For example, I thought your paragraph about the number of ethics consults is stronger than your other paragraphs in your discussion because you go into more detail about that there.

6. PLOS authors have the option to publish the peer review history of their article (what does this mean?). If published, this will include your full peer review and any attached files.

Reviewer #1: **Yes: **Kentaro Iwata

Reviewer #2: No

---

## [Author Response · Author response to Decision Letter 0]

13 Nov 2023

1. We thank the reviewer for their comments regarding the clarity of our aims, objectives, and hypotheses. The primary aim of this study was to examine the association between patient demographics and clinical characteristics with the timing and outcomes of ethics consultations. Our primary hypotheses consisted of the following: 

Non-English language speakers experience increased times from admission to ethics consultation. 

Differences in ethics consult timing and recommendations would be observed based on patient diagnosis. 

To address this comment, we have modified and added to the introduction of our manuscript to clarify the aims and objectives of the study. Additionally, the hypotheses tested and knowledge gaps addressed have been clearly defined in the last paragraph of our introduction. 

2.We thank the reviewer for this comment. In our study, outcomes refer to the timing of the ethics consultation from admission and the set of recommendations made by the ethics team. We did not directly measure patient outcomes like transition to end-of-life care. To address this comment, we have clarified this definition in our introductory paragraph. 

3. Thank you for raising this important point regarding the Cox regression used in the statistical analysis. Prior to performing the Cox regression, we calculated the standardized mean difference of each clinical or demographic variable between all language groups. The SMD analysis was performed to 1) determine whether demographic variables were different between language groups and may confound analyses of outcomes between language groups and 2) determine which variables to include in the multivariate Cox analysis of ethics consult timing between language groups. We would like to note that SMD has been previously used to find differences between groups as feature selection for this purpose. We found that the only demographic variable that was significantly different between language groups was age and therefore included this variable in the multivariate model. Other variables like sex were not significantly or meaningfully different between groups and were therefore excluded from the model. Please see supplemental table 1 of our manuscript for the variables that were significant and non-significant between groups. We found that after including Age in the cox model, we still observed significant differences in ethics consult timing between language groups, indicating that these differences are likely driven by language and are not confounded by other demographic features. 

4.Ethics consultations are conducted in a similar manner at our institution compared to the institutions at which ethics consultations have been previously studied. In the prior literature, ethics consultations were requested by physicians for a variety of reasons including conflict resolution, challenging interactions with patients or families, and to assist with clinical decision-making, notably end-of-life care. Similarly, ethics consultations at the institution where this study was conducted were requested by physicians for the same reasons as those listed above. While individual practices may differ by institution, namely the initiator of ethics consultations, the thorough process through which ethics consults were requested and conducted at our institution was overall similar to those documented in the single-center studies we cited (Spielman et al, Erler et al). The details of how ethics consultations were conducted in our study have been included in the “cohort identification” paragraph of the methods section. 

5.Thank you for highlighting this concern. Our primary and secondary outcomes were defined as the time from admission to ethics consultation and the recommendations made by the ethics consult team respectively. These definitions are detailed in the “chart review” paragraph of the methods section. 

6. In our study, patient demographics were defined based on self-reported patient information that was obtained during admission or prior outpatient visits at the institution. Recorded patient demographic information was based primarily on patient reported information and not the opinion of the recording provider or staff member. Unknown demographics are either not recorded or displayed in the chart as “Unknown”. We have clarified this information in the methods section of the text. 

7. In the study, primary diagnosis was defined as the primary reason for admission. These were obtained through chart review as ICD codes vary between provider and are inconsistent between patient groups. We have clarified the definition of primary diagnosis in the text and have included how this information was obtained in the “chart review” section of the methods. 

8. In our study, language fluency was determined by the patient’s reported preferred language. We have clarified this in the “chart review” section of the methods. Regarding the use of language aids, unfortunately, we are unable to verify or quantify the usage of translators as these were not consistently documented in the chart. We have included this limitation in the conclusion section of the manuscript. 

9.We have removed any content from our conclusion that was not directly related to the main objectives of our study which included ethics consult timing and recommendations in relation to demographic and clinical factors such as principal diagnosis. Specifically, we removed aspects of the conclusion related to the association between patient decision making capacity and ethics consult recommendations as this analysis revealed no significant differences between patients with and without capacity. We additionally removed discussion sections related to the number of ethics consultations performed in different groups in our cohort as the variability in this data was low and we did not identify significant differences between groups. Finally, we removed discussion sections related to the association between hospital location and ethics recommendation as these results were not related to our primary hypotheses. Please see below for further details and responses to other comments related to the conciseness of the discussion section. 

10. We thank the reviewer for their comment and agree that the reason for ethics consultation is important to evaluate across the cases in our study. We would like to note that since every ethics consult is individualized to the patient and their family, the reason for consultation was extremely variable across the cohort and each patient had a unique reason for consultation. We felt that categorization of this variable would diminish its importance and would not capture the variability present in the data. 

The reason for consultation is related to patients’ primary diagnoses, family dynamics, and team dynamics, among other variables. This was consistent with the prior literature we cited in this manuscript. We therefore did not categorize the reason for consultation into discrete groups. We note that some of the reasons for ethics consultation in our cohort included assisting with family discussions about patient care, determining patient’s surrogate decision maker, determining a patient’s candidacy for cesarian section, and assessing the benefit and appropriateness of a tracheostomy, among others. To address the reviewer’s comment, we have included this information about the reason for consultation in our results section. 

11. In the ethics team recommendations evaluated in our study, “transfer/letter” reflected a recommendation to transfer medical care to another institution or a direct transfer letter requesting further care at an alternative healthcare location. We have clarified this definition in the "chart review” section of the methods and in the legend of table 2. 

12. Thank you for raising this concern. We have added an x axis label to Figure 1A and have included p value labels in Figures 1A and 2. 

13. Thank you for this comment. The most important findings in our study were that patients admitted for COVID-19 had significantly longer median times to consult compared with other primary diagnoses and that Spanish-speaking patients had longer median times to consult than English-speaking patients. These findings have been included in the conclusions and abstract of our current manuscript. 

14. We have addressed all comments received from reviewers in the updated version of our manuscript. Please see below for responses to reviewer comments that were not already addressed above. 

15. We have reviewed and addressed all typographical and grammatical errors in the updated version of the manuscript. 

Reviewer 1: 

Major: 

We thank the reviewer for this comment and agree that assessment of patient-provider communication is important in patients lacking decisional capacity. At the institution where this study was performed, whenever possible, advance care directives were used in accordance with standard policy. If an advance directive was unavailable, a surrogate decision maker, usually a member of the patient’s family, was appointed for decision making. 

The reviewer raises an important point that the primary language of family members of patients lacking capacity may not reflect the patient’s spoken language. Unfortunately, the primary language of the family members was not consistently recorded in the EMR or notes in these instances. This is a limitation of our study which we have highlighted in the discussion section. 

Minor: 

We thank the reviewer for this helpful comment. We have ensured that each sentence reflects the proper tense. 

Reviewer 2: 

Introduction: 

We thank the reviewer for this insightful comment. While no literature exists quantifying the exact number of hospitalized patients with limited English proficiency, we cited a study quantifying limited English proficiency in the USA demonstrating that over 20 million individuals in the United States are limited English proficient. Furthermore, we have clarified our hypothesis to reflect that poor patient provider comprehension and poor health literacy in limited English proficient patients may lead to increased time to ethics consult, citing two related studies to justify this hypothesis. 

We thank the reviewer for this insightful comment. We hypothesized that different diagnoses would have variable ethics consult timing and recommendations due to differences in inpatient disease acuity, progression of disease upon admission, and reasons for the initiation of consultation. For example, we hypothesized that in patients with drawn out disease courses like COVID-19 and cancer, these patients would have ethics consults later in admission since these consults would be more likely to be centered around end-of-life care. We have included this reasoning in the introductory paragraphs and conclusion of the manuscript. 

Methods: 

Thank you for this recommendation. Regarding interactions between the ethics team and patient and/or family, some, but not all ethics consultations necessitate a meeting between ethics team members and the patient/family. In all cases, there is a meeting between members of the primary care team and ethics team to discuss recommendations for how to proceed with patient care. In all cases, the aim of the ethics team is to hold a meeting and provide recommendations within 24 hours of consultation. We have clarified this in the text. 

Under Chart Review: 

We thank the reviewer for this insightful comment. While our primary outcome of time to ethics consultation has been studied before, differences in this outcome between patients with different primary languages or diagnoses have not been assessed. To our knowledge, this is the first study to investigate the association between language spoken, patient diagnosis, and ethics consultation outcomes (time to consultation and recommendations). We have clarified this in the introduction, explicitly mentioning that “the association between patient language spoken, and ethics consult timing, center-wide as well as between different diagnoses has not been previously evaluated”. 

We thank the reviewer for this comment. While the frequency of ethics consultations by diagnosis is a knowledge gap, we acknowledge the reviewer’s additional point to consolidate the manuscript to focus on “a few findings (2-3) of interest”, focusing primarily on the associations between language, primary diagnosis, and ethics consult recommendations. Consistent with the recommendation to streamline the manuscript, we removed sections of our conclusions related to the frequency of ethics consultation in different groups in our cohort as the variability in this data was low and we did not identify significant differences between groups. 

Results: 

Please see our above response for the meaning of transfer/letter and clarification in the main text. Regarding Table 2, patients can be counted multiple times (one patient can have multiple recommendations). We have clarified this feature of the analysis in the Table 2 legend. 

Discussion: 

We thank the reviewer for this helpful comment. We have summarized the most important findings in the first paragraph of the discussion. In addition, we have added a sentence detailing how these findings add to what is already known in the literature, namely that understanding the association between language spoken and primary diagnosis can aid in understanding when to request an ethics consultation. Please see our responses to previous comments on details of the changes made to the discussion section. 

We thank the reviewer for this helpful comment. We have revised the first sentence of the discussion to specifically reflect that this is the first study to evaluate the association specifically between patient language spoken and primary clinical diagnosis in relation to patient outcomes. 

We thank the reviewer for this feedback. Given the prior reviewer comments regarding removing sections of the discussion for conciseness, we have removed the section about the number of ethics consults as these results consisted of negative data due to low n and low median number of ethics consults. 

We thank the reviewer for the helpful comment. In response to this comment and others, we have streamlined our discussion and bolstered aspects of the discussion which were lacking in detail. Please see the responses to prior comments for details on the major revisions made to the discussion section that address this comment.

---

## [Decision Letter · Decision Letter 1]

4 Dec 2023

PONE-D-23-13646R1Characterization of Patients Requiring Inpatient Hospital Ethics Consults- A Single Center StudyPLOS ONE

Dear Dr. Dayyani,

Thank you for submitting your manuscript to PLOS ONE. After careful consideration, we feel that it has merit but does not fully meet PLOS ONE’s publication criteria as it currently stands. Therefore, we invite you to submit a revised version of the manuscript that addresses the points raised during the review process.

1. Kindly reformat your Abstract to a 'structured abstract' with sections: "Introduction or Background, Methods, Results, Conclusions.2.  In the Methods Sections of the Body of the manuscript, "the variables to be included in the Cox proportional hazards model should not be predetermined or pre-mentioned, or you 'could could state that you used covariate with statistical significance by univariate analysis in the methods."3. Also, "use of covariate such as the languages or the age should only be included in the results."

We look forward to receiving your revised manuscript.

Kind regards,

Sylvester Chidi Chima, M.D., L.L.M.

Academic Editor

PLOS ONE

Journal Requirements:

Reviewers' comments:

Reviewer's Responses to Questions

**Comments to the Author**

1. If the authors have adequately addressed your comments raised in a previous round of review and you feel that this manuscript is now acceptable for publication, you may indicate that here to bypass the “Comments to the Author” section, enter your conflict of interest statement in the “Confidential to Editor” section, and submit your "Accept" recommendation.

Reviewer #1: All comments have been addressed

Reviewer #2: All comments have been addressed

2. Is the manuscript technically sound, and do the data support the conclusions?

Reviewer #1: Partly

Reviewer #2: Yes

3. Has the statistical analysis been performed appropriately and rigorously? 

Reviewer #1: No

Reviewer #2: Yes

4. Have the authors made all data underlying the findings in their manuscript fully available?

Reviewer #1: Yes

Reviewer #2: No

5. Is the manuscript presented in an intelligible fashion and written in standard English?

Reviewer #1: Yes

Reviewer #2: Yes

6. Review Comments to the Author

Reviewer #1: In the methods, the variables to be included in the Cox proportional hazards model should not be predetermined or pre-mentioned. The authors could state that they used covariate with statistical significance by univariate analysis in the methods, and use of covariate such as the languages or the age should only be included in the results.

Reviewer #2: The authors have satisfactorily assessed the comments by both reviewers. I recommend this paper for acceptance.

7. PLOS authors have the option to publish the peer review history of their article (what does this mean?). If published, this will include your full peer review and any attached files.

Reviewer #1: **Yes: **Kentaro Iwata

Reviewer #2: **Yes: **Gina Piscitello

---

## [Author Response · Author response to Decision Letter 1]

4 Dec 2023

1. Kindly reformat your Abstract to a 'structured abstract' with sections: "Introduction or Background, Methods, Results, Conclusions. 

We thank the editor for this comment and have structured our abstract to include the requested sections. 

2. In the Methods Sections of the Body of the manuscript, "the variables to be included in the Cox proportional hazards model should not be predetermined or pre-mentioned, or you 'could could state that you used covariate with statistical significance by univariate analysis in the methods." 

Thank you for this suggestion. We did not pre-select variables to include in our multivariate cox proportional hazards model. As previously stated in the rebuttal and summarized by the reviewer, we included features in the multivariate model based on the significance results of univariate testing. We have modified the language in our methods section to reflect that we performed covariate testing using the statistical significance of univariate analyses for selecting features. 

3. Also, "use of covariate such as the languages or the age should only be included in the results." 

We thank the reviewer for this suggestion and have modified the text to only mention the use of language and age as covariates in the results section of the manuscript.

---

## [Decision Letter · Decision Letter 2]

18 Dec 2023

Characterization of Patients Requiring Inpatient Hospital Ethics Consults- A Single Center Study

PONE-D-23-13646R2

Dear Dr. Dayyani,

We’re pleased to inform you that your manuscript has been judged scientifically suitable for publication and will be formally accepted for publication once it meets all outstanding technical requirements.

Kind regards,

Sylvester Chidi Chima, M.D., L.L.M, LLD.

Academic Editor

PLOS ONE

Reviewer's Responses to Questions

**Comments to the Author**

1. If the authors have adequately addressed your comments raised in a previous round of review and you feel that this manuscript is now acceptable for publication, you may indicate that here to bypass the “Comments to the Author” section, enter your conflict of interest statement in the “Confidential to Editor” section, and submit your "Accept" recommendation.

Reviewer #1: (No Response)

2. Is the manuscript technically sound, and do the data support the conclusions?

Reviewer #1: Yes

3. Has the statistical analysis been performed appropriately and rigorously? 

Reviewer #1: Yes

4. Have the authors made all data underlying the findings in their manuscript fully available?

Reviewer #1: Yes

5. Is the manuscript presented in an intelligible fashion and written in standard English?

Reviewer #1: Yes

6. Review Comments to the Author

Reviewer #1: No additional comment I have.

7. PLOS authors have the option to publish the peer review history of their article (what does this mean?). If published, this will include your full peer review and any attached files.

Reviewer #1: **Yes: **Kentaro Iwata

---

## [Editor Report · Acceptance letter]

24 Mar 2024

PONE-D-23-13646R2 

PLOS ONE

Dear Dr. Dayyani, 

I'm pleased to inform you that your manuscript has been deemed suitable for publication in PLOS ONE. Congratulations! Your manuscript is now being handed over to our production team.

Kind regards, 

on behalf of

Professor Sylvester Chidi Chima 

Academic Editor

PLOS ONE